# Perception and demand for healthy snacks/ beverages among US consumers vary by product, health benefit, and color

**Glory Esohe Okpiaifo** [ID][☯], **Bertille Dormoy-Smith** [ID][☯], **Bachir Kassas** [ID][☯], **Zhifeng Gao** [ID][☯]*

Food and Resource Economics Department, University of Florida, Gainesville, Florida, United States of America

☯ These authors contributed equally to this work.
* zfgao@ufl.edu

**Data Availability Statement:** All data files are available from Harvard Dataverse repository - https://doi.org/10.7910/DVN/V47JMW.

## Abstract

Concerns about the numerous health problems associated with unhealthy snacks prompted recommendations to steer individuals toward healthier eating habits. One such recommendation advises limiting unhealthy snacks and replacing them with more fruits and vegetables with significant health benefits. This study investigates US consumers' perceptions and preferences for healthy (vegetable-based) snacks/beverages. An online survey was designed to estimate consumer perception and willingness-to-pay (WTP) for vegetable-based crackers, spreads, and beverages. A sampling company sent the survey to its national consumer panels in 2020, resulting in a sample of 402 US consumers. Eligible participants were adults, primary grocery shoppers who consumed crackers, spreads, and beverages. Consumer WTP for healthy snacks/beverages, the dependent variable, was elicited using a payment card method. Independent variables include personality traits (Innovativeness and Extraversion) and the important factors affecting healthy snack purchases, health consciousness, and demographic variables. Results show that consumers' preferences for healthy snacking vary by product, even when the products have similar health benefits. Significant positive associations exist between WTP for healthy snacks/beverages and personality traits, health consciousness, and some demographics. This study provides critical insights to policymakers and informs marketing campaigns to promote healthy snacking in the US more effectively.

## Introduction

Snacking is a practice that is commonplace among people from all age groups, both in the US and the rest of the world [1]. Snacks can be described as food or caloric beverages consumed outside regular mealtimes [1]. Over 90% of adults in the United States reported consuming snacks at least once daily [2]. While the frequency of snacking has increased over time, consumption of fruits and vegetables has decreased [3, 4]. This raises significant health concerns since most snacks tend to be high in carbohydrates while providing minimal amounts of other essential nutrients [1]. High consumption of these unhealthy snacks (mainly salty snacks,

**Funding:** This study was supported by the Florida Agricultural Experiment Station and the USDA National Institute of Food and Agriculture (Hatch/ multistate project FLA-FRE-006196, [PI Zhifeng Gao]). The funders had no role in study design, data collection and analysis, decision to publish, or preparation of the manuscript.

**Competing interests:** The authors have declared that no competing interests exist.

desserts, candy, and sweetened beverages) would lead to an energy-dense and nutrient-poor diet [2]. Nutrient-poor diet associated with the consumption of unhealthy snacks substantially raises energy intake, leading to an increased prevalence of overweight and obesity, along with accompanying health problems, including type 2 diabetes, hypertension, dyslipidemia, and cardiovascular diseases [2, 5]. Negative associations have also been identified between unhealthy snacking, oral hygiene, satiety/appetite, and metabolism [6]. In addition, unhealthy snacking could impact the ability to maintain a balanced diet, contributing to poor eating habits and exacerbating weight gain [6, 7]. This is especially worse when unhealthy snacks are added to regular meal consumption without any compensation for the additional energy intake from these snacks. Therefore, there is a critical need to promote healthy snack options that would fulfill the snacking needs of individuals while providing a more nutrient-rich and calorie-balanced diet.

Healthy snacks can be described as snack choices that are more nutritionally balanced and intended to boost positive health outcomes. Examples of healthy snack choices are fruits, vegetables, and functional foods (consist of functional ingredients (which impart the health benefit) incorporated into a carrier/base food product, encompassing various food categories) that are modified to provide specific health benefits and reduce the risk of adverse health outcomes [8]. A growing awareness of the role of diet in improving wellbeing and life expectancy has led to a rise in consumer interest in healthy food/snack choices [9]. Despite the interest in healthy food/snack among consumers, most current research focuses on the consumption of unhealthy snacks. Research shows that the reasons for the prevalence of unhealthy snacks in diets vary by age group. Children and adults seem to base their choice of snacks on taste, thus biasing their choices towards unhealthy and indulgent snacks [10]. For college students, the stress of schoolwork, lack of time, and lack of experience and skills required to prepare healthy meals/snacks are primary factors underlying the predisposition for unhealthy snacking [6, 11]. Unemployment and socioeconomic status are also factors that may push adults towards unhealthy snacking due to a need to buy cheaper food and/or cope with psychological stress [6, 12]. It is therefore important to identify novel healthy snacking options that will be perceived favorably by consumers and fit their needs. Thus, this study aims to investigate consumers' preferences for multiple novel healthy snacking options.

There is a large body of work on consumer preferences for novel food technologies/products in general. In contrast to non-food domains, many technological innovations resulting in novel food products aren't perceived as favorably by consumers [13–15]. Given the importance of technological innovations to meeting global food demand, much research has been devoted to investigating the underlying factors driving these perceptions. Due to factors such as limited knowledge, consumers have been shown to utilize heuristics (e.g., emotions, trust, natural-is-better) in their evaluation of food products, which can result in biased decisions [13, 16]. The framing of food technology information can also influence consumers' acceptance of new products [17–19]. Additionally, food technology neophobia [20], disgust sensitivity [21], cultural differences [22], and personality traits such as openness and conscientiousness [23], may help explain some of the differences in consumer acceptance. Some other determinants revealed by various studies include food safety concerns [24], risk perceptions [24], socio-demographic factors (e.g., age, gender, education level) [25, 26], lifestyle habits (e.g., vegetarian, travelling habit) [25, 26]. Our study contributes to this literature by exploring preferences specifically for novel healthy snacking products.

As demand for healthy snacking choices continues to increase, more research is focusing on developing new products that can appeal to consumers. Recent work has examined consumer perceptions of novel functional products such as fortified farmed fish [27], granola bars

[28], enriched coffee [29], and probiotic yogurt [30], among others. Also, given the high nutritional value of vegetables and the growing global demand for plant-based food products [31], there is growing research into vegetable-based functional food product alternatives and general healthy snacking [32–34]. Research shows that when consumers become more aware of their health benefits, they report a significant increase in the demand for these foods [35, 36]. For example, one study [37] found that Turkish consumers had a favorable attitude towards functional food products, with a majority believing that these foods are necessary and part of a healthy diet. Other studies across various countries found similar results using outcomes such as willingness to try [38, 39], willingness to pay [40, 41], and willingness to buy/purchase intention [42–45]. These studies spanned various food and beverage products such as apples, tomatoes, yogurts, cereals, etc.

This stream of literature has identified a wide range of factors that possibly influence preferences for these novel products. Demographic factors such as age [27], gender [39], education level [46], and household size [46], and marital status [8] have been linked to willingness to consume various functional food products. The perceived healthiness of the product/ingredients [47], health information [41], knowledge of product brand [48], price [49], taste [45], and other product characteristics have also been found to influence preferences. Multiple studies have also reported associations between various psychological/behavioral characteristics such as, health consciousness [42], knowledge [50], trust [44], food neophobia [44], motivations [51], health-related behaviors [52], beliefs [52], and consumer preferences for novel food and beverage products.

Our study contributes to this growing literature and efforts to improve the nutrition quality of consumers' diets by examining consumer preferences for healthy snacking options and the possible predictors of these preferences. Specifically, this study investigates US consumer preferences for multiple novel healthy (vegetable-based) snacks, including crackers and spreads. Since sugar-sweetened beverages are a major source of calories for people who enjoy snacking [53], we also include vegetable-based beverages to examine how consumers' preferences differ for healthy snack foods vs. beverages. This study also examines several factors that influence consumer willingness-to-pay (WTP) for healthy snacks and beverages, including Innovativeness, Extraversion, and other behavioral and socioeconomic/demographic factors.

To the best of our knowledge, this is the first study to measure consumer WTP for multiple novel healthy snacks and beverage products that are differentiated by appearance and health benefits. The three focus product categories chosen for this study–crackers, spreads, and beverages–were selected because these are popular snacking options. This study provides valuable insights for food industries to develop more healthy snacks and beverages that appeal to consumers and inform marketing campaigns to promote healthy snacking. The results of this study can also benefit policymakers by providing information to guide dietary guidelines for healthy snacking recommendations.

## Methods

### Data collection

An online survey was conducted in 2020 using a nationally representative sample of 402 US consumers. Prior to the survey data collection, a focus group was conducted to collect feedback from participants that helped refine the survey instruments. The survey was also pretested multiple times to improve and finalize the questions. Eligible survey participants were at least 18 years old, primary grocery shoppers in the household, who consume crackers, spreads, and beverages other than water and milk. The survey duration was limited to 20 minutes to avoid respondent fatigue. The survey included attention check questions to improve data quality by

filtering out respondents who did not pay attention to the questions [54–56]. After providing consent and answering screening questions, respondents provided information relating to their snack and beverage consumption habits, attitudes towards health/environment, preferences for different healthy snack and beverage products, and behavioral and sociodemographic characteristics. The study was approved by the Institutional Review Board at the University of Florida (IRB 201901626), and participants provided written informed consent on the online questionnaire before answering the survey questions.

## WTP estimates

Respondents reported their preferences for six product alternatives in each of the three snack categories (crackers, spreads, and beverages). The products from each snack category were plant-based and produced using a zero-waste process, which was clearly explained to the respondents using text and an illustrative diagram (Fig 1). In this process, the vegetables are first pressed and fermented to produce the beverages. The vegetable byproducts from this step are then used to produce the crackers and spread, which would hold a similar taste profile and nutritional value to the beverages. The six products in each category were differentiated by color and health benefits (Table 1). Preferences were elicited using a payment card contingent valuation method (CVM). Before the valuation sections, the respondents were presented with the products' production process and health benefits, prices of related products in grocery stores, and a cheap talk script to control hypothetical bias. The respondents then proceeded to the payment card questions, where they provided their WTP for all six products in each product category (crackers, spreads, and beverages), totaling 18 WTP bids from each respondent.

CVM is a well-established method for eliciting stated preferences, which has gained traction in food marketing and behavioral economics [68, 69]. The payment card method is one of the CVMs in which respondents are presented with a series of prices (in ranges or point estimates) and asked to select the one that best reflects their maximum WTP. This method has been used to elicit WTP in several food studies [70–72]. In this study, we first ask respondents to select from a series of presented price ranges and then (based on their choice from the price ranges) ask them to further select from a series of presented price point estimates. This way, we elicit both interval and point WTP estimates. Example questions are illustrated in the S1 and S2 Appendices.

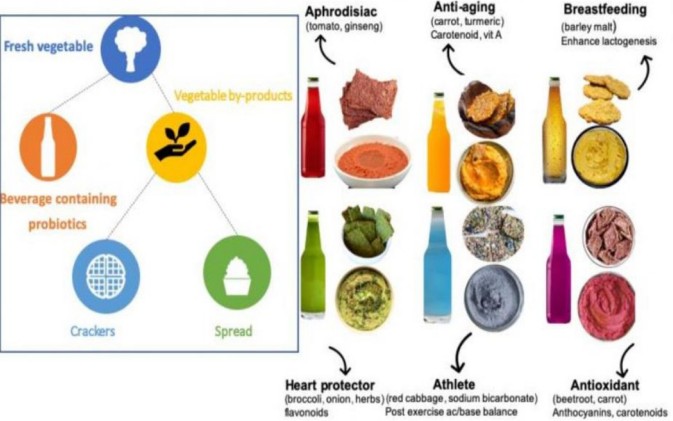

**Fig 1. Illustration of snack product alternatives and production process.**

Table 1. Description of snack product alternatives.

| Product | Composition | Health benefits |
|---|---|---|
| Red products | Tomato & Ginseng | Aphrodisiac [57, 58] |
| Orange products | Orange, Carrot, Turmeric | Anti-aging [59, 60] |
| Yellow products | Barley Malt | Aids breastfeeding [61] |
| Green products | Broccoli, Onion, Herbs | Heart protector [62–64] |
| Blue products | Red Cabbage, Sodium Bicarbonate | Improves athletic abilities [65, 66] |
| Purple products | Beetroot, Carrot | Antioxidant [60, 67] |

## Measurement of personality traits and behavioral variables

Psychology literature has established the importance of personality traits in humans, which, aside from demographics, capture the uniqueness of an individual. Personalities are usually expressed through thoughts, feelings, and actions. They are the foundation for numerous psychological and behavioral characteristics and have been extensively used to study various attitudes and behaviors in humans [73, 74]. Results from this stream of literature have linked personality traits to consumer decision-making and, more recently, consumer food-related choices [74, 75]. We thus explore the influence of two personality traits–Innovativeness and Extraversion–on consumer healthy snack and beverage preferences.

Innovativeness, a personality trait identified since the 1970s, describes an individual's willingness/ability to adopt new ideas before others. More specifically, consumer innovativeness relates to the tendency of an individual to purchase new products relatively earlier than other consumers [76]. Various scales have been used to measure consumer innovativeness, one of which is the Domain Specific Innovativeness Scale (DSI) developed by Goldsmith and Hofacker [77]. The DSI scale measures Innovativeness within a specific product category and has been employed in several food and non-food product studies [78–81]. Given the novelty of the focal products in this study, Innovativeness may influence respondents' WTP. For this reason, we utilize the DSI scale to measure the degree of respondents' Innovativeness regarding food. Respondents stated their agreement/disagreement for six statements (defined in Table 2)

Table 2. Definition of personality trait statements.

| Personality trait | Scoring |
|---|---|
| Innovativeness<br>• Compared to my friends, I purchase more new, different, or innovative food<br>• In general, I am amongst the first of my circle of friends to buy new, different, or innovative food.<br>• I buy new, different or innovative food before anyone else I know.<br>• Generally, I am amongst the first in my circle of friends to remember a brand of new, different or innovative food.<br>• If new, different, or innovative foods are available in shops and supermarkets I always purchase them.<br>• I do purchase new, different, or innovative foods even if I have not tasted/experienced them beforehand | 1 = Strongly disagree<br>2 = Disagree<br>3 = Somewhat disagree<br>4 = Neither agree nor disagree<br>5 = Somewhat agree<br>6 = Agree<br>7 = Strongly agree |
| Extraversion<br>• I enjoy human interaction<br>• I am enthusiastic<br>• I am talkative<br>• I am full of energy and I thrive on the presence of other people<br>• I take pleasure in activities that involve large social gathering<br>• I work well in a group<br>• I find few rewards in time spent alone<br>• I am bored when I am by myself | 1 = Strongly disagree<br>2 = Disagree<br>3 = Somewhat disagree<br>4 = Neither agree nor disagree<br>5 = Somewhat agree<br>6 = Agree<br>7 = Strongly agree |

on a 7-point Likert scale (1 = Strongly disagree, 7 = Strongly agree). Higher scores indicated a higher degree of Innovativeness.

The second personality trait we consider, Extraversion, relates to an individual's tendency to be friendly, sociable, and energetic. Extraverted consumers tend to be impulsive buyers [82], willing to purchase novel food products/try new aspects of food products [23, 75]. Also, since the food products used in this study are popular options at social gatherings, and extraverted individuals tend to be sociable, it is likely that their level of extraversion will be correlated with their consumption of these products. Based on these, the degree of Extraversion may influence respondents' willingness to purchase the products. We use the Myers-Briggs Type Indicator (MBTI), developed based on psychologist Carl Jung's theory of psychological type, to measure respondents' Extraversion [83]. The respondents stated their agreement/disagreement for the eight statements (defined in Table 2) on a 7-point Likert scale (1 = Strongly disagree, 7 = Strongly agree). Higher scores indicated higher Extraversion.

Other behavioral variables explored in this study include product-related factors (e.g., price, health benefits, flavor, non-GMO, organic, etc.) that respondents consider essential when purchasing healthy snack products. Given the health benefits and zero-waste nature of the products in this study, the health and environmental consciousness of the respondents may influence their preferences. The respondents were asked to rate statements of how much they care about their health and the environment on a 7-point Likert scale (1 = Strongly disagree, 7 = Strongly agree). They also rated the degree of the healthfulness of their diets on a scale from 1 (Very unhealthy) to 7 (Very healthy). Sociodemographic characteristics such as age, race, gender, presence of kids, household income, and education were also collected.

## Regression models

To determine the factors that influence WTP for each product, we estimated three Seemingly Unrelated Regressions (SUR), one for each product category. Due to the potential interrelatedness of subjects' preferences across the different alternatives in each product category, it is safe to assume that the error terms associated with the equations may be correlated. The SUR method is thus appropriate in this context since it allows for robustness of parameter covariance to error terms across the equations. Each regression has six linear equations, described below.

$$y_{ijk} = \alpha_n + \beta_{jk} X_{ijk} + e_{ijk}, i = 1 \dots N, j = 1 \dots 6, k = 1 \dots 3 \qquad (1)$$

where $y_{ijk}$ represents respondent $i$'s WTP for the $j$th product (*Red products*, *Orange products*, *Yellow products*, *Green products*, *Blue products*, or *Purple products*) in the $k$th category (*crackers*, *spread*, or *beverages*), $X_{ijk}$ represents the vector of selected independent variables and $e_{ijk}$ represents the unobserved variables. For each regression, the dependent variables were different based on the product differentiated by color and health benefits, while the independent variables were the same for all six equations. The selected independent variables include Innovativeness, Extraversion, important factors that influence the purchase of healthy drinks/snacks (Health benefits, Flavor, Non-GMO, Organic, Price), health consciousness (healthfulness of food consumed, care for health), and demographics. In the final model, environmental consciousness was not included because it was statistically insignificant in all regression models.

## Results

### Summary statistics

Table 3 presents a summary of the sociodemographic and behavioral characteristics of the sample. Approximately 55% are females, and the majority (87.6%) are Caucasian/White.

**Table 3. Summary of demographic and behavioral characteristics (n = 402).**

| Variable | % | Variable | % |
|---|---|---|---|
| *Gender* | | *Income* | |
| Male | 45 | 1 = Under $14,999 | 8.21 |
| Female | 55 | 2 = $15,000-$24,999 | 10.20 |
| *Age* | | 3 = $25,000-$34,999 | 12.94 |
| 1 = 18–24 | 8.46 | 4 = $35,000-$49,999 | 8.96 |
| 2 = 25–34 | 18.66 | 5 = $50,000-$74,999 | 19.15 |
| 3 = 35–44 | 19.15 | 6 = $75,000-$99,999 | 12.94 |
| 4 = 45–54 | 16.92 | 7 = $100,000-$149,999 | 13.93 |
| 5 = 55–64 | 17.41 | 8 = $150,000-$199,999 | 7.21 |
| 6 = 65–74 | 14.18 | 9 = > = $200,000 | 6.47 |
| 7 = > = 75 | 5.22 | | |
| *Education* | | *Race* | |
| 1 = Less than high school | 0.50 | 1 = Hispanic | 7.46 |
| 2 = Some high school/high school graduate | 16.42 | 2 = White | 87.59 |
| 3 = Some college, no degree | 16.17 | 3 = Black/African American | 6.84 |
| 4 = Associate's degree, occupational/academic | 12.19 | 4 = Asian | 3.54 |
| 5 = Bachelor's degree | 28.86 | 5 = Others | 2.03 |
| 6 = Master's degree | 20.65 | | |
| 7 = Professional/Doctoral degree | 5.22 | | |
| *Marital Status* | | | |
| 1 = Single/never married | 23.94 | | |
| 2 = Co-habiting | 4.74 | | |
| 3 = Married | 53.87 | | |
| 4 = Widowed | 5.74 | | |
| 5 = Divorced/Separated | 11.72 | | |
| | | Mean | Median | Std. Dev. |
| Extraversion | | 37.30 | 38 | 9.15 |
| Innovativeness | | 26.58 | 28 | 9.99 |
| Care for health | | 6.28 | 7 | 0.96 |
| Care for environment | | 5.84 | 6 | 1.22 |

Around half the subjects are married (53.9%) and have a Bachelor's degree or higher (54.7%). Subjects seem to be evenly distributed across the different age and income categories. Regarding the behavioral factors, average Innovativeness and Extraversion scores were 26.58 and 37.30, respectively. With maximum scores of 42 for Innovativeness and 56 for Extraversion, our results indicate a relatively high degree of Extraversion and Innovativeness among our sample. The average scores for health and environmental consciousness were 6.28 and 5.84, respectively, indicating a health and environmentally conscious sample.

## Consumption frequency/importance of health benefits

We evaluated how frequently the respondents consume various food products and beverages (Fig 2). The consumption frequencies for all food products are high, including snacks. Consumption frequency was relatively higher for water, tea/coffee, milk, soda, fruit juice, and lower for sport drinks, energy drinks, alternative milk, and vegetable drinks. We also asked the respondents to rate the importance of six health benefits that can be provided by snacks/

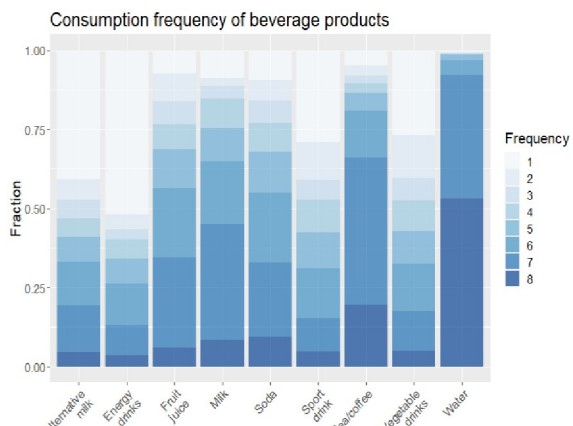

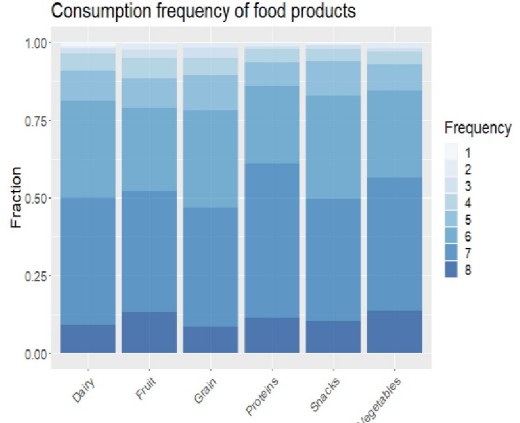

**Fig 2. Consumption frequency of food and beverage products.** *Notes*: Frequency: 1 –Almost never. 2 –Several times a year, but not every month. 3 –Monthly. 4- Several times a month, but not every week. 5- Weekly. 6- Several times a week, but not every day. 7 –Daily. 8 –More than once per day.

beverages in this study when purchasing food products (Fig 3). The graph shows that the heart protector and antioxidant health benefits are very important to the respondents, while the least important health benefits are Aphrodisiac and Breastfeeding.

## Important factors that influence the purchase of healthy snacks/beverages

Fig 4 presents the importance of factors that influence respondents' purchase of healthy products (snacks/beverages). The respondents seem to have similar preferences for both snacks and beverages across most factors. Flavor is the most important factor that respondents consider when purchasing healthy snacks and beverages, followed by price. The least important factor seems to be organic.

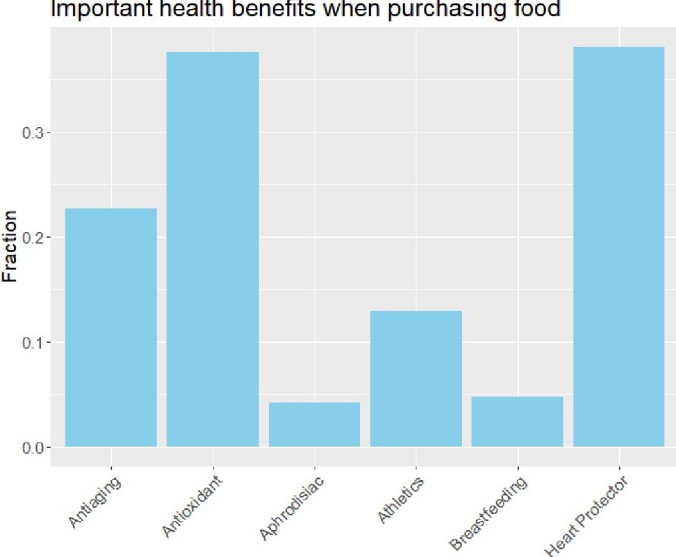

**Fig 3. Important health benefits when purchasing food products.** *Notes*: Each bar represents the fraction of the respondents that selected the corresponding health benefit.

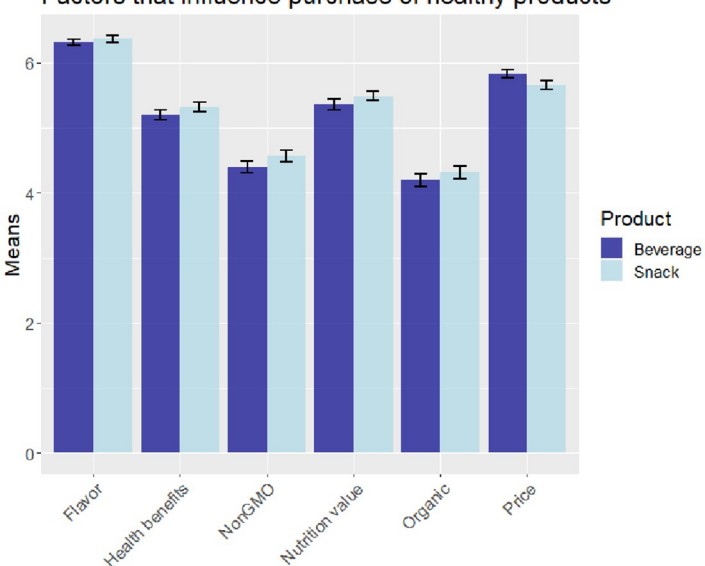

**Fig 4. Important factors that influence the purchase of healthy beverages/snacks.** *Notes*: Y-axis represents mean ratings. Ratings—1 = Extremely unimportant. 2 = Unimportant. 3 = Slightly unimportant. 4 = Neither important nor unimportant. 5 = Slightly important. 6 = Important. 7 = Extremely important.

## Willingness to pay

The average WTP for each product is presented in Fig 5. First, we found that the WTP for all alternatives in all three product categories are positive and significant. This shows that consumers do indeed have significant preferences for healthy snacks/beverages. Importantly, in all 3 product categories, pairwise tests (Wilcoxon signed rank tests) show that the WTP for the

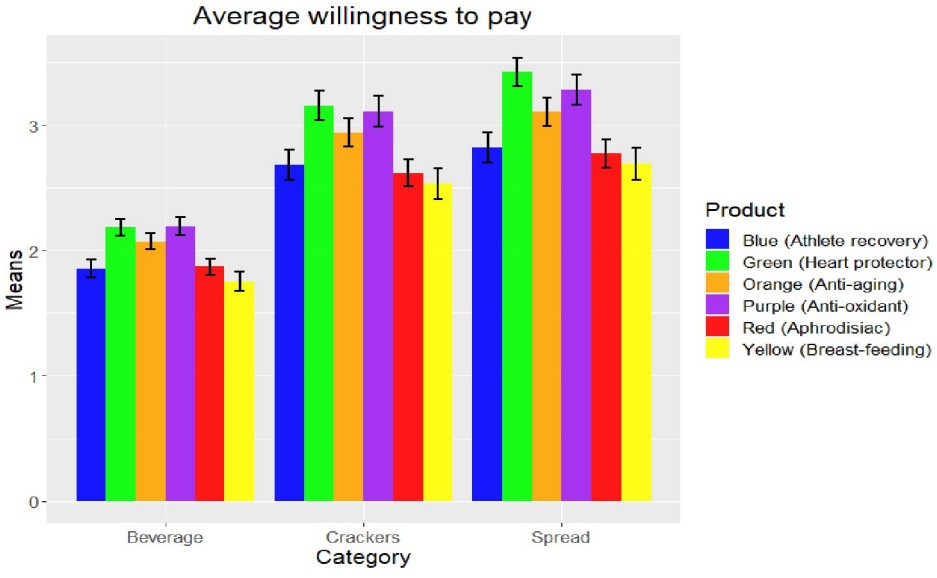

**Fig 5. Average WTP bids.** *Notes*: Friedman test was used to test for significant differences between products in each category (p<0.000).

green (heart protector) and the purple (antioxidant) products are significantly higher (p<0.09 for crackers, p<0.01 for spreads, p<0.05 for beverages) than the other four alternatives. This is an interesting result and seems to corroborate the earlier results showing that respondents place higher importance on heart protector and antioxidant health benefits than on the other health benefits. The average WTP for green products are $2.18, $3.16 and $3.43, respectively, for beverages, crackers, and spreads. For purple products, average WTP are $2.19, $3.11, and $3.28, respectively, for beverages, crackers, and spreads. Friedman test results indicate statistically significant differences in WTP between the products within each category (p<0.000), thus confirming that consumer preferences for healthy snacks/beverages vary by health benefits.

Results from the SUR model over beverage products are presented in Table 4. We observe a significant positive influence of Innovativeness on WTP for orange (anti-aging) and purple (antioxidant) drinks, indicating that respondents who are more likely to try new things (i.e., have a higher innovativeness score) are willing to pay more for these drinks. Extraversion is also positively correlated with WTP for red (aphrodisiac), green (heart-protector), and yellow (breastfeeding) drinks, indicating that individuals who are more friendly/sociable hold higher WTP for these drinks. We also find that individuals who consider health benefits as a primary factor when purchasing healthy drinks are willing to pay more for all the beverage options except for red (aphrodisiac) and yellow (breastfeeding). In contrast, the perceived importance of price seems to drive a negative effect on WTP for the orange (anti-aging) beverage, implying that consumers who place high importance on price are willing to pay less for the orange beverage. The stronger the respondents' impression of the healthfulness of their diet, the higher their WTP for all beverage options except the green (heart-protector) and orange (anti-aging) drinks. Looking at demographics in Table 4, we see a significant negative correlation between age and WTP for all beverage products, indicating that younger individuals are willing to pay more for the healthy beverages. We also find evidence that African Americans have a higher WTP for the red (aphrodisiac) and yellow (breastfeeding) drinks, Asians have a lower WTP for the purple (antioxidant) drink, and those with kids have a higher WTP for the green (heart-protector) drink.

Results from the SUR model over the crackers product alternatives are presented in Table 5. Innovativeness is positively correlated with the WTP for all crackers except for the green (heart-protector) alternative, indicating that people with a higher willingness to try new things (i.e., have a higher innovativeness score) are willing to pay more for these products. People who consider flavor as an important factor when purchasing healthy snacks are willing to pay more for the red (aphrodisiac) crackers only. In contrast, people who pay attention to organic are willing to pay more for the green (heart-protector) crackers. The stronger the respondents' impression of the healthfulness of their food consumption, the higher their WTP for every cracker product alternative. Looking at demographic variables in Table 5, we see a significant negative correlation between age and WTP for all crackers. We also observe that Caucasians are willing to pay more for the red (aphrodisiac), yellow (breastfeeding), and green (heart-protector) crackers, while African Americans are willing to pay more for the yellow (breastfeeding) crackers only. Increasing household income is associated with higher WTP for all crackers except blue (athlete recovery) and purple (antioxidant).

Results from the SUR model over the spread products are presented in Table 6. We observe a significant positive influence of Innovativeness on the WTP for just the blue (athlete recovery) and purple (antioxidant) spreads, indicating that people who are willing to try new things (i.e., higher innovativeness scores) are willing to pay more for these spreads. Extraversion is positively correlated with the WTP for the yellow (breastfeeding) spread only, indicating that individuals who are more friendly/sociable (i.e., have a higher extraversion score) are willing

**Table 4. Willingness-to-pay for healthy drink regression.**

|  | Red (Aphro-rodisiac) | Orange (Anti-aging) | Yellow (Breast feeding) | Green (Heart protector) | Blue (Athlete recovery) | Purple (Anti- oxidant) |
|---|---|---|---|---|---|---|
| Intercept | 0.364 (0.685) | 0.672 (0.714) | 0.578 (0.799) | 1.222 (0.768) | 0.914 (0.751) | 1.148 (0.792) |
| Innovativeness | 0.009 (0.008) | 0.017** (0.008) | 0.009 (0.010) | 0.007 (0.009) | 0.014 (0.009) | 0.028*** (0.009) |
| Extraversion | 0.016** (0.008) | 0.012 (0.008) | 0.026*** (0.009) | 0.023*** (0.009) | 0.014 (0.009) | 0.008 (0.009) |
| **Important factors that influence the purchase of healthy drinks** |  |  |  |  |  |  |
| Health benefits | -0.004 (0.048) | 0.087* (0.050) | -0.039 (0.056) | 0.165*** (0.054) | 0.115** (0.053) | 0.117** (0.056) |
| Flavor | 0.063 (0.071) | 0.057 (0.074) | -0.109 (0.083) | -0.103 (0.080) | -0.011 (0.078) | -0.079 (0.082) |
| Non-GMO | 0.009 (0.044) | 0.009 (0.046) | 0.072 (0.051) | -0.028 (0.049) | 0.010 (0.048) | -0.035 (0.051) |
| Organic | 0.034 (0.048) | -0.019 (0.050) | -0.014 (0.056) | -0.006 (0.054) | -0.011 (0.053) | -0.058 (0.056) |
| Price | -0.075 (0.053) | -0.128** (0.056) | 0.041 (0.062) | -0.084 (0.060) | -0.098* (0.058) | -0.065 (0.062) |
| **Health variables** |  |  |  |  |  |  |
| Healthfulness of food consumed | 0.127*** (0.050) | -0.001 (0.052) | 0.126** (0.059) | 0.057 (0.056) | 0.136*** (0.055) | 0.111* (0.058) |
| Care for health | -0.046 (0.066) | 0.106 (0.069) | -0.003 (0.077) | 0.050 (0.074) | -0.026 (0.072) | 0.083 (0.076) |
| **Demographics** |  |  |  |  |  |  |
| Age | -0.138*** (0.042) | -0.102** (0.043) | -0.175*** (00048) | -0.142*** (0.047) | -0.207*** (0.046) | -0.121*** (0.048) |
| Hispanic | 0.013 (0.243) | -0.015 (0.254) | -0.463 (0.284) | -0.428 (0.273) | 0.133 (0.267) | -0.186 (0.281) |
| Caucasian | 0.512 (0.351) | -0.107 (0.366) | 0.203 (0.410) | 0.221 (0.394) | 0.289 (0.385) | -0.253 (0.406) |
| African-American | 0.667* (0.387) | -0.101 (0.403) | 0.797* (0.451) | 0.189 (0.433) | 0.546 (0.424) | -0.065 (0.447) |
| Asian | -0.216 (0.447) | -0.105 (0.466) | -0.382 (0.522) | -0.094 (0.501) | -0.647 (0.490) | -0.855* (0.517) |
| Female | 0.052 (0.129) | -0.054 (0.134) | -0.231 (0.150) | 0.080 (0.144) | -0.120 (0.141) | 0.074 (0.149) |
| Married | 0.183 (0.146) | 0.139 (0.152) | 0.252 (0.170) | 0.047 (0.163) | 0.114 (0.160) | 0.155 (0.168) |
| Presence of kids | -0.070 (0.163) | 0.096 (0.170) | 0.208 (0.190) | 0.314* (0.182) | 0.121 (0.178) | 0.255 (0.188) |
| Household income | 0.033 (0.038) | 0.051 (0.040) | 0.018 (0.044) | 0.040 (0.042) | 0.001 (0.042) | 0.026 (0.044) |
| Education | -0.014 (0.051) | 0.002 (0.053) | 0.011 (0.059) | -0.051 (0.057) | 0.006 (0.056) | 0.002 (0.059) |

*Notes*: N = 402, OLS $R^2$ = 0.25.

* 10% significance level.

** 5% significance level.

*** 1% significance level.

Numbers in parentheses are standard errors.

**Table 5. Willingness-to-pay for healthy crackers regression.**

| | Red (Aphro-rodisiac) | Orange (Anti-aging) | Yellow (Breastfeeding) | Green (Heart protector) | Blue (Athlete recovery) | Purple (Anti- oxidant) |
|---|---|---|---|---|---|---|
| Intercept | -2.490** (1.225) | -1.250 (1.280) | -0.954 (1.342) | -1.279 (1.294) | 0.279 (1.324) | -1.044 (1.369) |
| Innovativeness | 0.032** (0.014) | 0.027* (0.015) | 0.031** (0.015) | 0.024 (0.015) | 0.041*** (0.015) | 0.034** (0.016) |
| Extraversion | 0.008 (0.014) | 0.004 (0.014) | 0.020 (0.015) | 0.021 (0.014) | 0.021 (0.015) | 0.020 (0.015) |
| **Important factors that influence purchase of healthy snacks** | | | | | | |
| Health benefits | -0.078 (0.088) | -0.099 (0.092) | -0.098 (0.096) | -0.046 (0.093) | -0.078 (0.095) | -0.072 (0.098) |
| Flavor | 0.291*** (0.118) | 0.197 (0.123) | 0.101 (0.129) | 0.134 (0.125) | -0.008 (0.128) | 0.111 (0.132) |
| Non-GMO | 0.007 (0.072) | 0.014 (0.076) | 0.038 (0.079) | -0.017 (0.076) | 0.049 (0.078) | 0.102 (0.081) |
| Organic | 0.069 (0.081) | 0.079 (0.085) | 0.039 (0.089) | 0.144* (0.086) | -0.005 (0.087) | 0.006 (0.090) |
| Price | -0.023 (0.081) | -0.027 (0.084) | -0.032 (0.088) | -0.083 (0.085) | -0.047 (0.087) | -0.141 (0.090) |
| **Health variables** | | | | | | |
| Healthfulness of food consumed | 0.226*** (0.089) | 0.278*** (0.093) | 0.270*** (0.098) | 0.232*** (0.094) | 0.342*** (0.096) | 0.281*** (0.099) |
| Care for health | 0.002 (0.113) | -0.037 (0.118) | 0.006 (0.124) | -0.029 (0.120) | -0.054 (0.123) | 0.060 (0.127) |
| **Demographics** | | | | | | |
| Age | -0.241*** (0.072) | -0.201*** (0.075) | -0.352*** (0.079) | -0.187*** (0.076) | -0.381*** (0.078) | -0.254*** (0.080) |
| Hispanic | 0.268 (0.421) | 0.433 (0.440) | 0.265 (0.461) | 0.401 (0.445) | 0.061 (0.455) | 0.198 (0.470) |
| Caucasian | 1.460** (0.606) | 0.768 (0.633) | 1.149* (0.664) | 1.212* (0.640) | 0.675 (0.655) | 0.812 (0.677) |
| African-American | 0.515 (0.669) | 0.285 (0.699) | 1.256* (0.733) | 0.713 (0.707) | 0.627 (0.723) | 0.027 (0.747) |
| Asian | 0.483 (0.773) | 0.000 (0.808) | 0.282 (0.847) | 0.758 (0.816) | -0.563 (0.835) | -0.480 (0.864) |
| Female | 0.088 (0.222) | 0.104 (0.232) | -0.181 (0.243) | 0.262 (0.234) | 0.115 (0.240) | 0.156 (0.248) |
| Married | -0.292 (0.252) | -0.033 (0.263) | -0.151 (0.276) | 0.219 (0.266) | 0.134 (0.272) | 0.056 (0.281) |
| Presence of kids | 0.413 (0.282) | -0.097 (0.295) | 0.382 (0.309) | 0.222 (0.298) | 0.189 (0.305) | 0.517 (0.315) |
| Household income | 0.200*** (0.065) | 0.216*** (0.068) | 0.154** (0.072) | 0.138** (0.069) | 0.106 (0.071) | 0.117 (0.073) |
| Education | -0.059 (0.088) | 0.025 (0.092) | -0.047 (0.097) | -0.038 (0.093) | -0.056 (0.095) | 0.030 (0.099) |

*Notes*: N = 402, OLS $R^2$ = 0.25.

* 10% significance level.

** 5% significance level.

*** 1% significance level.

**Table 6. Willingness-to-pay for healthy spread regression.**

| | Red (Aphro-rodisiac) | Orange (Anti-aging) | Yellow (Breastfeeding) | Green (Heart protector) | Blue (Athlete recovery) | Purple (Anti- oxidant) |
|---|---|---|---|---|---|---|
| Intercept | -2.658** (1.303) | -1.880 (1.275) | -1.548 (1.343) | -0.808 (1.320) | 0.381 (1.350) | -1.016 (1.331) |
| Innovativeness | 0.0214 (0.015) | 0.012 (0.015) | 0.019 (0.015) | 0.004 (0.015) | 0.029* (0.015) | 0.032** (0.015) |
| Extraversion | 0.002 (0.015) | 0.009 (0.014) | 0.029** (0.015) | 0.018 (0.015) | 0.005 (0.015) | 0.011 (0.015) |
| **Important factors that influence purchase of healthy snacks** | | | | | | |
| Health benefits | 0.024 (0.093) | 0.139 (0.091) | -0.057 (0.096) | 0.116 (0.094) | -0.030 (0.096) | 0.065 (0.095) |
| Flavor | 0.381*** (0.126) | 0.159 (0.123) | 0.111 (0.129) | 0.110 (0.127) | -0.077 (0.130) | 0.170 (0.128) |
| Non-GMO | -0.026 (0.077) | -0.001 (0.075) | 0.101 (0.079) | -0.001 (0.078) | 0.054 (0.080) | 0.018 (0.078) |
| Organic | 0.067 (0.086) | 0.016 (0.084) | 0.003 (0.089) | 0.062 (0.087) | 0.025 (0.089) | -0.007 (0.088) |
| Price | -0.103 (0.086) | -0.133 (0.084) | -0.032 (0.089) | 0.062 (0.087) | -0.035 (0.089) | -0.128 (0.088) |
| **Health variables** | | | | | | |
| Healthfulness of food consumed | 0.239*** (0.095) | 0.226** (0.093) | 0.274*** (0.098) | 0.204** (0.096) | 0.270*** (0.098) | 0.261*** (0.097) |
| Care for health | -0.017 (0.121) | 0.043 (0.118) | -0.075 (0.124) | -0.011 (0.122) | 0.033 (0.125) | -0.055 (0.123) |
| **Demographics** | | | | | | |
| Age | -0.168** (0.076) | -0.156** (0.075) | -0.293*** (0.079) | -0.160** (0.077) | -0.321*** (0.079) | -0.140* (0.078) |
| Hispanic | 0.306 (0.448) | 0.772* (0.438) | 0.344 (0.462) | 0.291 (0.454) | 0.130 (0.464) | 0.081 (0.457) |
| Caucasian | 1.299** (0.645) | 1.236** (0.631) | 1.364** (0.664) | 1.248* (0.653) | 0.997 (0.668) | 0.777 (0.658) |
| African-American | 1.282* (0.712) | 1.358** (0.696) | 1.561** (0.733) | 1.061 (0.721) | 1.162 (0.737) | 0.869 (0.727) |
| Asian | 0.368 (0.822) | 0.524 (0.804) | 0.620 (0.847) | 0.525 (0.833) | -0.354 (0.851) | -0.459 (0.840) |
| Female | -0.097 (0.236) | -0.116 (0.231) | -0.510** (0.243) | -0.037 (0.239) | -0.141 (0.244) | -0.156 (0.241) |
| Married | -0.355 (0.268) | -0.055 (0.262) | -0.201 (0.276) | -0.161 (0.271) | -0.178 (0.277) | -0.082 (0.274) |
| Presence of kids | 0.273 (0.300) | -0.088 (0.294) | 0.649** (0.309) | 0.631** (0.304) | 0.529* (0.311) | 0.529* (0.307) |
| Household income | 0.209*** (0.070) | 0.197*** (0.068) | 0.187*** (0.072) | 0.188*** (0.071) | 0.158** (0.072) | 0.237*** (0.071) |
| Education | 0.012 (00094) | 0.075 (0.092) | 0.026 (0.097) | -0.017 (0.095) | -0.005 (0.097) | -0.015 (0.096) |

*Notes*: N = 402, OLS $R^2$ = 0.25.

* 10% significance level.

** 5% significance level.

*** 1% significance level.

to pay more for this spread. People who place importance on flavor when purchasing healthy snacks are willing to pay more for the red (aphrodisiac) spread, and the stronger the respondents' impression of the healthfulness of their food consumption, the higher their WTP for all spreads. Looking at demographic variables in Table 6, we see that younger respondents are willing to pay more for all the spreads. We also find suggestive evidence that Hispanics are willing to pay more for the orange (anti-aging) spread, while Caucasians are willing to pay more for all spreads except blue (athlete recovery) and purple (antioxidant), and African Americans are willing to pay more for the red (aphrodisiac), orange (anti-aging) and yellow (breastfeeding) spreads. We find a significant negative correlation between females and WTP for the yellow (breastfeeding) spread, while people with kids are willing to pay more for the yellow (breastfeeding), green (heart protector), blue (athlete recovery), and purple (antioxidant) spreads. Income is also positively correlated with WTP for all spreads. Together, results from Tables 4–6 confirm hypothesis (iii) by showing significant correlations between WTP for healthy snacks/beverages and several behavioral and sociodemographic factors.

## Discussion

The prevalence of unhealthy snacking, which has been associated with negative health outcomes, underscores the need to promote healthier snacking alternatives that appeal to consumers. This study evaluated consumer preferences for healthy crackers, spreads, and drinks, which are differentiated by color and health benefits. Out of six health benefits presented in the different product alternatives, respondents seem most attracted to two–heart protector and antioxidant–and are willing to pay higher premiums for the products carrying these benefits. A possible explanation for this result is that respondents are more familiar with these health benefits and/or perceive them as more primary/important than the other health benefits presented. Since the products in this study are related to functional food products, the results can be discussed in the context of functional foods literature. Previous work on consumer acceptance of functional food shows that familiarity with the health benefits provided by functional foods will boost consumer acceptance [84]. Although consumers have been shown to be more willing to purchase functional foods based on their health benefits [85], our study shows that consumers also differentiate between different health benefits, preferring some over others. In line with our results, earlier studies of functional food found that purchase intention is higher when the product's health claims are physiological (e.g., less risk of cardiovascular disease) rather than psychological [36, 43, 86]. For instance, information about reduced cholesterol benefits increased consumers' purchase intention for a fortified yogurt drink [87]. In another study, information about antioxidant benefits increased consumers' purchase intention for functional foods [41].

Understanding factors underlying consumer preferences for healthy products will help marketers and policymakers to promote healthier snacking behavior. In our study, we highlight the relationship between various personality/sociodemographic/behavioral factors and consumer WTP for the healthy snack products. The influence of psychological characteristics on consumer decision-making for food products is an important topic that remains relatively underexplored, specifically in the area of preference for healthy snacking products. Some studies have examined the significance of psychological factors such as perceptions, beliefs, attitudes, trust and food neophobia [50, 51, 88–94]. Our results show that Innovativeness is one factor that exhibits a strong relationship with WTP for multiple products across the three product categories, which is reasonable considering the novelty of the products used in the study. For example, purple is an unusual color for a drink, cracker, or spread, and "anti-aging" is not a popular health benefit marketed in food products. Our results conform with the

literature showing that Innovativeness predicts willingness to try various new healthy food/drink products [8, 95, 96]. Innovative consumers are a very important group because they are willing to try new products first and are usually the first to spread information about these products to others [8]. Consumers are influenced by these innovative consumers when making decisions about purchasing new products [97]. Given this background and our results, marketing campaigns for new healthy food products should first target innovative consumers.

Extraversion is also significantly correlated with WTP for some beverage and spread products. For example, extraverted individuals prefer red (aphrodisiac), green (heart-protector), and yellow (breastfeeding) beverages, showing that personality traits may also influence consumers' differentiation of health benefits. This is consistent with previous research results, which showed that extroverted consumers are more willing to purchase novel aspects of food products [23, 98]. The confirmed positive correlations between Innovativeness/Extraversion and preferences for various products can be used by marketers to explore personality-based marketing. Research shows that this type of marketing strategy can be done using platforms such as social media and is effective [99, 100].

Consumers consider several factors (extrinsic and intrinsic product characteristics) as they decide to purchase healthy food products. We explored the influence of some of these factors on WTP for healthy snacks/beverages. Health benefit shows up as a strong predictor of WTP for most beverage products. This is in line with the literature showing consumers are attracted to healthy products based on their health benefits [39, 44, 85, 101]. This is especially beneficial to marketers if consumers are familiar with the health benefits [84]. Thus, marketing campaigns could increase consumers' awareness and knowledge of various health benefits of healthy snacks and use labels to display these attributes more clearly on the products. The respondents' perceived importance of price seems to exert a small negative influence on only two of the beverage products (Orange, Anti-aging, and Blue, Athlete recovery), which could imply that these two benefits are not as important to the respondents' beverage purchase. For the rest of the products, however, the respondents are not deterred by price. This conforms with Huang et al. [49] and Ares et al. [101], who argue that functional food consumers are usually more interested in the health benefits of the products than the price. Flavor strongly influences WTP for the red (aphrodisiac) crackers and spreads only. The strong positive impact of flavor is in line with previous sensory studies that identified flavor as a critical attribute contributing to consumer acceptance of functional food [102, 103]. The result that flavor was only correlated with the WTP for the red crackers and spread is interesting and could be explained by the fact that consumers placed the lowest importance on the health benefit of the red products (i.e., aphrodisiac). Respondents who place higher importance on organic have a higher WTP for the green (heart protector) crackers. This is a reasonable result given the importance of this health benefit to the respondents, and considering that individuals who prefer organic foods tend to be health conscious [104, 105]. The positive effect of consumers' health consciousness, and attention to the healthiness of their diet, on acceptance of healthy food products, is well-established in the literature [45, 49, 106]. These studies can explain the strong positive correlation between the respondents' perception of the healthiness of their diets and their WTP for almost all products in the study.

Our results show a significant influence of various demographics on WTP for health snacks, confirming the conclusion of previous research on healthy food consumption. We find a negative correlation between age and WTP for all products used in our study. This is in line with some earlier studies that reported a higher interest in functional foods among younger individuals [37, 107]. However, it also contradicts the results of several past studies, where older individuals were found to have a higher WTP for healthy food products compared to younger individuals [50, 108–111]. One possible explanation is that older individuals are not convinced

by/do not trust the health claims of the snacks, or they do not believe that snacks can be healthy. Previous studies found that in contrast to younger individuals, older individuals are more attracted to healthy food products when the health claims refer to a reduction of negative outcomes (e.g., reduction of risk for heart disease) rather than health benefits [112, 113]. Another possible reason could be due to the novelty of the products since younger individuals tend to be open-minded and willing to try novel foods [96]. Race was also correlated with the WTP for some snack products, with the results showing differences, particularly in preferences among Caucasians and African Americans. This could guide marketing campaigns to target the specific needs of certain consumer groups. We do not find a significant correlation between gender and WTP for most of the products, which is contrary to previous findings of higher acceptance of healthy food products among females compared to males [50, 111, 114, 115]. Our result could be driven by females' skepticism about the health claims of the novel products in this study. We also find a higher WTP for people with kids across some of the products, which can be explained by the fact that parents of young children tend to be nutrition conscious, and they tend to pay attention to the healthiness of the foods they purchase, to ensure that their kids are in good health [116, 117]. Lastly, household income strongly influences WTP for all spreads and most of the crackers. This is similar to other studies, which found that higher income consumers have higher purchase intentions for healthy food products [29, 107, 118–120].

Previous functional foods studies found that consumers care not just about the ingredients but also about the functional food carrier/base product used [38, 49, 93, 121, 122]. Some products are perceived by consumers to be intrinsically healthier/more credible and thus preferred [85, 123]. This can explain the key differences between our results for the beverages, crackers, and spreads. For example, Innovativeness seems to be an important characteristic in all three product categories, but most especially for the crackers, as it influences the WTP for all the cracker product alternatives besides the green (heart protector) crackers. On the other hand, it only affects WTP for two product alternatives from beverages and spreads. A possible reason for this is that the novelty of the crackers is more appealing to the respondents than the beverages/spreads. The exception of the green crackers may be because the respondents do not perceive green crackers' characteristics (color/health benefit/ingredients) as novel as those of other alternatives. Another notable difference is that Extraversion has no effect on crackers, a small effect on one of the spreads, but a stronger effect on beverages. Also, the health benefits of the beverages seem to be more important to respondents than those of the snacks/spreads. A possible explanation is that the respondents perceive the beverages to be healthier carriers of the ingredients than the crackers and spreads. In contrast, for the crackers and spreads, flavor is the attribute considered more important. Another interesting difference is that respondents who tend to consume healthier food, in general, are willing to pay more for all the alternatives in all three product categories, except for the orange (anti-aging) and green (heart protector) beverages. One key finding is that household income was not significantly correlated with WTP for the beverages but positively correlated with WTP for crackers and spreads, which implies the differential effect of income on health snack consumption that are product dependent.

## Conclusion/implications for research and practice

Our results show that US consumers have strong preferences for plant-based healthy snacking. Their preferences for healthy snacking vary by product category and the specific health benefit that can be provided by the products. In addition, several key personality, demographic, and attitudinal variables have significant effects on consumer WTP for healthy snacks. However,

the effect of these variables also varies depending on the health benefits and the product category. Results uncovered in this study provide useful and critical information for the food industry/policymaker to produce/promote healthy food products that are more likely to succeed in the marketplace. Future research may estimate the relationship between healthy snack consumption and health outcomes. In addition, identifying the messages that can be used to educate consumers more effectively about healthy snacks is critical to developing effective education programs among different consumer groups to promote healthy food consumption.

Finally, despite the useful insights afforded in this study, it also has some limitations. First, consumer preferences were elicited using a stated preference approach, where the respondents reported their valuation in a hypothetical setting. This was necessary in this study since the products being investigated do not yet exist in the market. While we included a cheap talk script to improve the accuracy of the stated preferences, hypothetical bias is still possible (where respondents report higher valuations than they would be willing to pay in a real shopping scenario). Future studies could investigate this point further by comparing preferences for similar products in a hypothetical vs. incentive-compatible setting. Second, our sample size is relatively small for a nationwide survey. Although the current sample size ensures meaningful statistical power for all analyses conducted in this study, it may not support in-depth sub-analyses over different consumer groups that vary in socioeconomic or behavioral characteristics. Future research could investigate these sub-analyses to potentially uncover important insights that could help identify segments of the market.

## Supporting information

**S1 Appendix. Example valuation question (point estimate).**
(TIF)

**S2 Appendix. Example valuation question (interval estimate).**
(TIF)

## Author Contributions

**Conceptualization:** Bertille Dormoy-Smith.

**Formal analysis:** Glory Esohe Okpiaifo.

**Investigation:** Bertille Dormoy-Smith, Bachir Kassas, Zhifeng Gao.

**Methodology:** Bertille Dormoy-Smith, Bachir Kassas, Zhifeng Gao.

**Writing – original draft:** Glory Esohe Okpiaifo.

**Writing – review & editing:** Bachir Kassas, Zhifeng Gao.

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
