## [Decision Letter · Decision Letter 0]

14 Feb 2023

PONE-D-22-31581Perception and Demand for Healthy Snacks/Beverages among US Consumers Vary by Product, Health Benefit, and ColorPLOS ONE

Dear Dr. Gao,

Thank you for submitting your manuscript to PLOS ONE. After careful consideration, we feel that it has merit but does not fully meet PLOS ONE’s publication criteria as it currently stands. Therefore, we invite you to submit a revised version of the manuscript that addresses the points raised during the review process.

ACADEMIC EDITOR: Please check the details provided below.

We look forward to receiving your revised manuscript.

Kind regards,

Charles Odilichukwu R Okpala

Academic Editor

PLOS ONE

Journal Requirements:

2. Please provide additional details regarding participant consent. In the ethics statement in the Methods and online submission information, please ensure that you have specified what type you obtained (for instance, written or verbal, and if verbal, how it was documented and witnessed). If your study included minors, state whether you obtained consent from parents or guardians.

3. Please upload a new copy of Figures 1, 2, 3, 4 and 5 as the detail is not clear. Please follow the link for more information: " ext-link-type="uri" xlink:type="simple">https://blogs.plos.org/plos/2019/06/looking-good-tips-for-creating-your-plos-figures-graphics/"
https://blogs.plos.org/plos/2019/06/looking-good-tips-for-creating-your-plos-figures-graphics

Additional Editor Comments :

Please authors, kindly attend to the comments raised by the reviewers.

Kindly pay attention to the justification of this current study, this needs further details. Why is this study relevant?

Also, the methodology of the work, please provide additional details.

Also, kindly strengthen the discussion with additional relevant literature.

Look forward to your revised manuscript.

Reviewers' comments:

Reviewer's Responses to Questions

**Comments to the Author**

1. Is the manuscript technically sound, and do the data support the conclusions?

Reviewer #1: Yes

Reviewer #2: Yes

2. Has the statistical analysis been performed appropriately and rigorously? 

Reviewer #1: I Don't Know

Reviewer #2: Yes

3. Have the authors made all data underlying the findings in their manuscript fully available?

Reviewer #1: Yes

Reviewer #2: Yes

4. Is the manuscript presented in an intelligible fashion and written in standard English?

Reviewer #1: Yes

Reviewer #2: Yes

5. Review Comments to the Author

Reviewer #1: The manuscript "Perception and Demand for Healthy Snacks/Beverages among US Consumers Vary by Product,

Health Benefit, and Color" is a good piece of literature and can be recommended for publication in PLOS ONE after major revisions.

I've some suggestions for its improvement;

1) The authors have presented a graphical abstract, I don't think the journal requires a structured abstract, please confirm.

2) The structure of introduction is really confusing, authors have presented a mix information in this section. i.e., in first paragraph authors are giving a background and in second paragraph they are giving the objectives of the study while in 3rd paragraph again the authors have presented details about the topic. The objectives of the study and conceptualization should be shared at end of introduction.

3) The quality of figures provided is really poor and its being very difficult to interpret the results.

4) Authors should provide high quality illustration

5) Authors should add a graphical abstract

6) The conclusion part is missing in the main manuscript file

Reviewer #2: This study investigated consumer preferences for healthy crackers, spreads, and drinks, which are differentiated by color and health benefits. To address their research question, the authors estimated the willingness to pay for the afore mentioned products. Overall, the manuscript is well structured and amazingly written.

The research question is visibly defined, and an informative literature review is presented to support it. In addition, the authors clearly described how the instruments were administered to the subjects, and the approach is reliable according to the published literature. Based on the variables studied, appropriate statistical tests were applied. Lastly, the implications for research and practice are supported by the data.

What I adored the most about this manuscript and its findings is the fact that US consumers are willing to pay for functional foods that are manufactured with zero waste. This will prompt hope in achieving sustainability.

Recommended Minor revisions:

1. Discussion section may be improved by displaying more data already existing in the literature and compare them with those obtained in this study.

2. In the conclusions, I recommend including the limitations observed while undertaking this investigation.

6. PLOS authors have the option to publish the peer review history of their article (what does this mean?). If published, this will include your full peer review and any attached files.

Reviewer #1: No

Reviewer #2: No

---

## [Author Response · Author response to Decision Letter 0]

15 Mar 2023

Response to Academic Editor and Reviewers

Response to Academic Editor

Comment 1: Please ensure that your manuscript meets PLOS ONE's style requirements, including those for file naming. The PLOS ONE style templates can be found at

Response to comment 1: Thank you for your comment. We have formatted the title page and the manuscript headings to meet PLOS ONE’s style requirements.

Comment 2: Please provide additional details regarding participant consent. In the ethics statement in the Methods and online submission information, please ensure that you have specified what type you obtained (for instance, written or verbal, and if verbal, how it was documented and witnessed). If your study included minors, state whether you obtained consent from parents or guardians.

Response to comment 2: Thank you for your comment. We have specified the consent type obtained.

Line 104-109

“The study was approved by the Institutional Review Board at the University of Florida (IRB 201901626), and participants were provided written informed consent before answering the survey questions. The last part of the informed consent read as, “By choosing "continue" on the question below, you are indicating that you voluntarily agree to participate in this survey.” And respondents who did not select “continue” would exit the survey.”

Comment 3: Please upload a new copy of Figures 1, 2, 3, 4 and 5 as the detail is not clear. Please follow the link for more information: https://blogs.plos.org/plos/2019/06/looking-good-tips-for-creating-your-plos-figures-graphics/

Response to comment 3: Thank you for your comment. We have made new copies of the figures using the PLOS ONE tool. 

Additional Editor comments

Comment 4: Please authors, kindly attend to the comments raised by the reviewers.

Response to comment 4: Thank you. We have addressed all the reviewers’ comments and included details below on revisions made to address each comment raised by each reviewer.

Comment 5: Kindly pay attention to the justification of this current study, this needs further details. Why is this study relevant?

Response to comment 5: Thank you for your comment. Given the prevalence of unhealthy snacks and their associated negative health outcomes, our study contributes to efforts to identify healthy snacking options that will be appealing to consumers and economically viable. We explained this justification in the manuscript by giving a background on the problem – unhealthy snacking – and why studies like ours are needed. 

To make sure that this relevance is clearer and to address your comments, we added a statement addressing our contribution to literature and efforts to improve nutrition quality.

Lines 74 – 76

“Our study contributes to this growing literature and efforts to improve nutrition quality by examining consumers’ preferences for healthy snacking options and the possible predictors of these preferences.”

Lines 84 - 91 also provide more information on our contribution to the literature and food industry/policy makers –

“To the best of our knowledge, this is the first study to measure consumer WTP for multiple novel healthy snacks and beverage products that are differentiated by appearance and health benefits. The three focus product categories chosen for this study – crackers, spreads and beverages – were selected because these are popular snacking options at social gatherings. This study provides valuable insights for food industries to develop more healthy snacks and beverages that appeal to consumers and inform marketing campaigns to promote healthy snacking. The results of this study can also benefit policymakers by providing information to guide dietary guidelines of healthy snacking recommendations.” 

Comment 6: Also, the methodology of the work, please provide additional details. 

Response to comment 6: Thank you for your comment. We have provided additional details about the consent type, the survey progression, and also included a table defining the statements used to capture the personality traits.

Lines 104-109

“The study was approved by the Institutional Review Board at the University of Florida (IRB 201901626), and participants were provided written informed consent before answering the survey questions. The last part of the informed consent read as, “By choosing "continue" on the question below, you are indicating that you voluntarily agree to participate in this survey.” And respondents who did not select “continue” would exit the survey.”

Line 101-105

“After providing consent and answering screening questions, respondents provided information relating to their snack and beverage consumption habits, attitudes towards health/environment, preferences for different healthy snack and beverage products, and behavioral and sociodemographic characteristics”

Lines 169-172

Table 2. Definition of personality trait statements

Personality trait

Innovativeness

- Compared to my friends, I purchase more new, different, or innovative food

- In general, I am amongst the first of my circle of friends to buy new, different, or innovative food.

- I buy new, different or innovative food before anyone else I know.

- Generally, I am amongst the first in my circle of friends to remember a brand of new, different or innovative food.

- If new, different, or innovative foods are available in shops and supermarkets I always purchase them.

- I do purchase new, different, or innovative foods even if I have not tasted/experienced them beforehand

Extraversion

- I enjoy human interaction

- I am enthusiastic

- I am talkative

- I am full of energy and I thrive on the presence of other people

- I take pleasure in activities that involve large social gathering

- I work well in a group

- I find few rewards in time spent alone

- I am bored when I am by myself

Note: Respondents’ agreement/disagreement with these statements was measured using the five-point Likert scale: 1= Strongly disagree, 2=Disagree, 3= Somewhat disagree, 4= Neither agree nor disagree, 5= Somewhat agree, 6=Agree, 7= Strongly agree

Comment 7: Also, kindly strengthen the discussion with additional relevant literature.

Response to comment 7: Thank you for your comment. We have added more text discussing the results in the context of previous studies, and also included additional citations throughout the discussion. For example – 

Lines 341-346

“In line with our results, earlier studies of functional food found that purchase intention is higher when the health claims of the product are physiological (e.g., less risk of cardiovascular disease) rather than psychological (52–54). For instance, information about reduced cholesterol benefits resulted in an increase in consumers purchase intention for a fortified yoghurt drink (55), and in another study, information about antioxidants benefits increased consumers purchase intention for functional foods (56).”

Lines 352-356

“Some studies have examined the significance of psychological factors such as perceptions, beliefs, attitudes, trust and food neophobia (57–65). Our results show that Innovativeness is one factor that exhibits a strong relationship with the WTP for multiple products across the three product categories, which is reasonable considering the novelty of the products used in the study.”

Lines 384-387

“For the rest of the products, however, the respondents are not deterred by price. This conforms with Huang et al. (75) and Ares et al. (73) who argue that functional food consumers are usually more interested in the health benefits of the products than the price.”

Response to Reviewer 1

Comment 1: The authors have presented a graphical abstract, I don't think the journal requires a structured abstract, please confirm.

Response to comment 1: Thank you for your comment. We rewrote the abstract to meet PLOS ONE’s requirements.

Comment 2: The structure of introduction is really confusing, authors have presented a mix information in this section. i.e., in first paragraph authors are giving a background and in second paragraph they are giving the objectives of the study while in 3rd paragraph again the authors have presented details about the topic. The objectives of the study and conceptualization should be shared at end of introduction.

Response to comment 2: Thank you for your comment, and apologies for any confusion in the introduction. Following your suggestion, we have restructured the introduction and moved the objectives paragraph next to the contributions at the end of the section. 

Comment 3: The quality of figures provided is really poor and its being very difficult to interpret the results.

Response to comment 3: Thank you for your comment. We have made new copies of the figures using the PLOS ONE tool, which significantly improved the quality of the figures.

Comment 4: Authors should provide high quality illustration

Response to comment 4: Thank you for your comment. As mentioned in the response to your third comment, we have made new copies of the figures using the PLOS ONE tool and this significantly improved the quality of the illustrations in the manuscript. 

Comment 5: Authors should add a graphical abstract

Response to comment 5: Thank you but we are confused by this comment. In your comment 1, you said that “The authors have presented a graphical abstract….” but in this comment, you ask us to add a graphical abstract. We are really confused. 

But anyways, based on your comment 1, we have provided an abstract that meets PLOS ONE requirements. 

Comment 6: The conclusion part is missing in the main manuscript file

Response to comment 6: Thank you for your comment. The “Implications for Research and Practice” section was meant to serve as the conclusion. We have renamed this section to avoid confusion, and we also edited this section with more information about limitations and suggestions for future work.

Line 448

“Conclusion/Implications for Research and Practice”

Line 460-472

“Finally, despite the useful insights afforded in this study, it also has some limitations. First, consumer preferences were elicited using a stated preference approach, where the respondents reported their valuation in a hypothetical setting. This was necessary in this study since the products being investigated do not yet exist in the market. While we included a cheap talk script to improve the accuracy of the stated preferences, there is still a possibility of hypothetical bias occurring (where respondents report higher valuation than they would be willing to pay in a real shopping scenario). Future studies could investigate this point further by comparing preferences for similar products in a hypothetical vs incentive compatible setting. Second, our sample size is relatively small for a nationwide survey. Although the current sample size ensures meaningful statistical power for all analyses conducted in this study, it may not support in-depth sub-analyses over different consumer groups that vary in socioeconomic or behavioral characteristics. Future research could investigate these sub-analyses to potentially uncover important insights that could help identify segments of the market.”

Response to Reviewer 2

Comment 1: Discussion section may be improved by displaying more data already existing in the literature and compare them with those obtained in this study.

Response to comment 1: Thank you for your comment. We have added more text discussing the results in the context of previous studies, and also included additional citations throughout the discussion. For example – 

Lines 341-346

“In line with our results, earlier studies of functional food found that purchase intention is higher when the health claims of the product are physiological (e.g., less risk of cardiovascular disease) rather than psychological (52–54). For instance, information about reduced cholesterol benefits resulted in an increase in consumers purchase intention for a fortified yoghurt drink (55), and in another study, information about antioxidants benefits increased consumers purchase intention for functional foods (56).”

Lines 352-356

“Some studies have examined the significance of psychological factors such as perceptions, beliefs, attitudes, trust and food neophobia (57–65). Our results show that Innovativeness is one factor that exhibits a strong relationship with the WTP for multiple products across the three product categories, which is reasonable considering the novelty of the products used in the study.”

Lines 384-387

“For the rest of the products, however, the respondents are not deterred by price. This conforms with Huang et al. (75) and Ares et al. (73) who argue that functional food consumers are usually more interested in the health benefits of the products than the price.”

Comment 2: In the conclusions, I recommend including the limitations observed while undertaking this investigation.

Response to comment 2: Thank you for your comments. We have included a paragraph on limitations

Line 460-472

“Finally, despite the useful insights afforded in this study, it also has some limitations. First, consumer preferences were elicited using a stated preference approach, where the respondents reported their valuation in a hypothetical setting. This was necessary in this study since the products being investigated do not yet exist in the market. While we included a cheap talk script to improve the accuracy of the stated preferences, there is still a possibility of hypothetical bias occurring (where respondents report higher valuation than they would be willing to pay in a real shopping scenario). Future studies could investigate this point further by comparing preferences for similar products in a hypothetical vs incentive compatible setting. Second, our sample size is relatively small for a nationwide survey. Although the current sample size ensures meaningful statistical power for all analyses conducted in this study, it may not support in-depth sub-analyses over different consumer groups that vary in socioeconomic or behavioral characteristics. Future research could investigate these sub-analyses to potentially uncover important insights that could help identify segments of the market.”

---

## [Decision Letter · Decision Letter 1]

10 Apr 2023

PONE-D-22-31581R1Perception and Demand for Healthy Snacks/Beverages among U.S. Consumers Vary by Product, Health Benefit, and ColorPLOS ONE

Dear Dr. Gao,

Thank you for submitting your manuscript to PLOS ONE. After careful consideration, we feel that it has merit but does not fully meet PLOS ONE’s publication criteria as it currently stands. Therefore, we invite you to submit a revised version of the manuscript that addresses the points raised during the review process.

ACADEMIC EDITOR: Please see below.

We look forward to receiving your revised manuscript.

Kind regards,

Charles Odilichukwu R. Okpala

Academic Editor

PLOS ONE

Journal Requirements:

Additional Editor Comments:

Please, kindly address the comments raised by the reviewer, it will help improve the quality of this work.

Reviewers' comments:

Reviewer's Responses to Questions

**Comments to the Author**

1. If the authors have adequately addressed your comments raised in a previous round of review and you feel that this manuscript is now acceptable for publication, you may indicate that here to bypass the “Comments to the Author” section, enter your conflict of interest statement in the “Confidential to Editor” section, and submit your "Accept" recommendation.

Reviewer #2: All comments have been addressed

Reviewer #3: (No Response)

2. Is the manuscript technically sound, and do the data support the conclusions?

Reviewer #2: Yes

Reviewer #3: (No Response)

3. Has the statistical analysis been performed appropriately and rigorously? 

Reviewer #2: Yes

Reviewer #3: (No Response)

4. Have the authors made all data underlying the findings in their manuscript fully available?

Reviewer #2: Yes

Reviewer #3: (No Response)

5. Is the manuscript presented in an intelligible fashion and written in standard English?

Reviewer #2: Yes

Reviewer #3: (No Response)

6. Review Comments to the Author

Reviewer #2: My comments have been critically addressed by the author.

The manuscript's quality has been elevated.

Reviewer #3: In general, the reviewer's comments have been addressed well in the revised version. However, the introduction part must be improved: The flow of the introduction is not clear, why the study is needed and why it is important. Most of the introduction is about improving healthy snacks, not about consumer perception. While the study is investigating the consumers feedback. I would suggest focusing on consumer perceptions. For example, what is the current trend in the perception of potential consumers about developing new products (in general)? Then continue with the importance of healthy snacks, and continue to emphasize the current aim.

7. PLOS authors have the option to publish the peer review history of their article (what does this mean?). If published, this will include your full peer review and any attached files.

Reviewer #2: **Yes: **Mouandhe Imamou Hassani

Reviewer #3: No

---

## [Author Response · Author response to Decision Letter 1]

23 May 2023

Response to Academic Editor and Reviewers

Response to Academic Editor

Comment 1: Please, kindly address the comments raised by the reviewer, it will help improve the quality of this work. 

Response to comment 1: Thank you for your comment. We have addressed the reviewer’s comments and included details below on revisions made to address the comments

Response to Reviewer 3

Comment: In general, the reviewer’s comments have been addressed well in the revised version. However, the introduction part must be improved: The flow of the introduction is not clear, why the study is needed and why it is important. Most of the introduction is about improving healthy snacks, not about consumer perception. While the study is investigating the consumers feedback. I would suggest focusing on consumer perceptions. For example, what is the current trend in the perception of potential consumers about developing new products (in general)? Then continue with the importance of healthy snacks, and continue to emphasize the current aim. 

Response to comment: Thank you for your comment. We have incorporated your suggestions to improve the introduction section by adding discussions about consumer perceptions of new products (in general) and novel snacking products. We also reorganized the introduction to make the flow better. 

Lines 71 – 109

There is a large body of work on consumer preferences for novel food technologies/products in general. In contrast to non-food domains, many technological innovations resulting in novel food products aren’t perceived as favorably by consumers (39–41). Given the importance of technological innovations to meeting global food demand, much research has been devoted to investigating the underlying factors driving these perceptions. Due to factors such as limited knowledge, consumers have been shown to utilize heuristics (e.g., emotions, trust, natural-is-better) in their evaluation of food products, which can result in biased decisions (39,42). The framing of food technology information can also influence consumers’ acceptance of new products (43–45). Additionally, food technology neophobia (46), disgust sensitivity (47), cultural differences (48), and personality traits such as openness and conscientiousness (49), may help explain some of the differences in consumer acceptance. Some other determinants revealed by various studies include food safety concerns (50), risk perceptions (50), socio-demographic factors (e.g., age, gender, education level) (51,52), lifestyle habits (e.g., vegetarian, travelling habit) (51,52). Our study contributes to this literature by exploring preferences specifically for novel healthy snacking products.

As demand for healthy snacking choices continues to increase, more research is focusing on developing new products that can appeal to consumers. Recent work has examined consumer perceptions of novel functional products such as fortified farmed fish (13), granola bars (14), enriched coffee (15), and probiotic yogurt (16), among others. Also, given the high nutritional value of vegetables and the growing global demand for plant-based food products (17), there is growing research into vegetable-based functional food product alternatives and general healthy snacking (18–20). Research shows that when consumers become more aware of their health benefits, they report a significant increase in the demand for these foods (21,22). For example, one study (23) found that Turkish consumers had a favorable attitude towards functional food products, with a majority believing that these foods are necessary and part of a healthy diet. Other studies across various countries found similar results using outcomes such as willingness to try (24,25), willingness to pay (26,27), and willingness to buy/purchase intention (28–31). These studies spanned various food and beverage products such as apples, tomatoes, yogurts, cereals, etc. 

This stream of literature has identified a wide range of factors that possibly influence preferences for these novel products. Demographic factors such as age (13), gender (25), education level (32), and household size (32), and marital status (8) have been linked to willingness to consume various functional food products. The perceived healthiness of the product/ingredients (33), health information (27), knowledge of product brand (34), price (35), taste (31), and other product characteristics have also been found to influence preferences. Multiple studies have also reported associations between various psychological/behavioral characteristics such as, health consciousness (28), knowledge (36), trust (30), food neophobia (30), motivations (37), health-related behaviors (38), beliefs (38), and consumer preferences for novel food and beverage products

---

## [Editor Report · Decision Letter 2]

2 Jun 2023

Perception and Demand for Healthy Snacks/Beverages among U.S. Consumers Vary by Product, Health Benefit, and Color

PONE-D-22-31581R2

Dear Dr. Gao,

We’re pleased to inform you that your manuscript has been judged scientifically suitable for publication and will be formally accepted for publication once it meets all outstanding technical requirements.

Kind regards,

Charles Odilichukwu R. Okpala

Academic Editor

PLOS ONE

Additional Editor Comments (optional):

The editor is very satisfied with the revised version. It is acceptable for publication.
---

## [Editor Report · Acceptance letter]

8 Jun 2023

PONE-D-22-31581R2 

Perception and Demand for Healthy Snacks/Beverages among US Consumers Vary by Product, Health Benefit, and Color 

Dear Dr. Gao:

I'm pleased to inform you that your manuscript has been deemed suitable for publication in PLOS ONE. Congratulations! Your manuscript is now with our production department. 

Kind regards, 

on behalf of

Dr. Charles Odilichukwu R. Okpala 

Academic Editor

PLOS ONE